# Predictive auxiliary objectives in deep RL mimic learning in the brain

## Abstract

The ability to predict upcoming events has been hypothesized to comprise a key aspect of natural and machine cognition. This is supported by trends in deep reinforcement learning (RL), where self-supervised auxiliary objectives such as prediction are widely used to support representation learning and improve task performance. Here, we study the effects predictive auxiliary objectives have on representation learning across different modules of an RL system and how these mimic representational changes observed in the brain. We find that predictive objectives improve and stabilize learning particularly in resource-limited architectures. We identify settings where longer predictive horizons better support representational transfer. Furthermore, we find that representational changes in this RL system bear a striking resemblance to changes in neural activity observed in the brain across various experiments. Specifically, we draw a connection between the auxiliary predictive model of the RL system and hippocampus, an area thought to learn a predictive model to support memory-guided behavior. We also connect the encoder network and the value learning network of the RL system to visual cortex and striatum in the brain, respectively. This work demonstrates how representation learning in deep RL systems can provide an interpretable framework for modeling multi-region interactions in the brain. The deep RL perspective taken here also suggests an additional role of the hippocampus in the brain– that of an auxiliary learning system that benefits representation learning in other regions.

## 1 Introduction

Deep reinforcement learning (RL) models have shown remarkable success solving challenging problems (Sutton & Barto, 2018; Mnih et al., 2013; Silver et al., 2016; Schulman et al., 2017). These models use neural networks to learn state representations that support complex value functions. A key challenge in this setting is to avoid degenerate representations that support only subpar policies or fail to transfer to related tasks. Self-supervised auxiliary objectives, particularly predictive objectives, have been shown to regularize learning in neural networks to prevent overfit or collapsed representations (Lyle et al., 2021; Dabney et al., 2021; François-Lavet et al., 2019). As such, it is common to combine deep RL objectives with auxiliary objectives. The modular structure of these multi-objective models can function as a metaphor for how different regions of the brain combine to comprise an expressive, generalizable learning system.

Analogies can readily be drawn between the components of a deep RL system augmented with predictive objectives and neural counterparts. For instance, the striatum has been identified as a RL-like value learning system (Schultz et al., 1997). Hippocampus has been linked to learning predictive models and cognitive maps (Mehta et al., 1997; O'Keefe & Nadel, 1978; Koene et al., 2003). Finally, sensory cortex has been suggested to undergo unsupervised or self-supervised learning akin to feature learning (Zhuang et al., 2021), although reward-selective tuning also been observed (Poort et al., 2015). It is unclear how value learning, predictive objectives, and feature learning mutually interact to shape representations. Comparing representations across artificial and biological neural networks can provide a useful frame of reference for understanding the extent artificial models resemble the brain's mechanisms for robust and flexible learning.

These comparisons can also provide useful insights into neuroscience, where little is known about how learning in one region might drive representational changes across the brain. For instance, the hippocampus is a likely candidate for predictive objectives, as ample experimental evidence has

shown that activity in this region is predictive of the upcoming experience of an animal (Skaggs & McNaughton, 1996; Lisman & Redish, 2009; Mehta et al., 1997; Payne et al., 2021; Muller & Kubie, 1989; Pfeiffer & Foster, 2013; Schapiro et al., 2016; Blum & Abbott, 1996; Mehta et al., 2000). These observations are often accounted for in theoretical work as hippocampus computing a predictive model or map (Lisman & Redish, 2009; Mehta et al., 2000; Russek et al., 2017; Whittington et al., 2020; Momennejad, 2020; George et al., 2021; Stachenfeld et al., 2017). Much has been written about how learned predictive models may be used by the brain to simulate different outcomes or support planning (Vikbladh et al., 2019; Geerts et al., 2020; Mattar & Daw, 2018; Miller et al., 2017; Ólafsdóttir et al., 2018; Redish, 2016; Koene et al., 2003; Foster & Knierim, 2012; McNamee et al., 2021). However, in the context of deep RL, the mere act of learning to make predictions in one region confers substantial benefits to other interconnected regions by shaping representations to incorporate predictive information (Hamrick et al., 2020; Oord et al., 2018; Bengio, 2012). One of the key insights of this work is to propose that an additional role of predictive learning in hippocampus is to drive representation learning that supports deep RL in the brain.

The main contribution of this paper is to quantify how representations in a deep RL model change with predictive auxiliary objectives, and to identify how these changes mimic representational changes in the brain. We first characterize functional benefits this auxiliary system confers on learning. We evaluate the effects of predictive auxiliary objectives in a simple gridworld foraging task, and confirm that these objectives help prevent representational collapse, particularly in resource-limited networks. We also observe that longer-horizon predictive objectives are more useful than shorter ones for transfer learning. We further demonstrate that a deep RL model with multiple objectives undergo a variety of representational phenomena also observed in neural populations in the brain. Downstream objectives can alter activity in the encoder, which is mirrored in various results that show how visual cortical activity is altered by both predictive and value learning. Additionally, learning in the prediction module drives activity patterns consistent with activity measured in hippocampus. Overall we find that interacting objectives explain diverse effects in the neural data not well modeled by considering learning systems in isolation. Moreover, this suggests that deep RL with predictive objectives appears to in many ways mirror the brain's approach to learning.

## 2 RELATED WORK

In deep RL, auxiliary objectives have emerged as a crucial tool for representation learning. These additional objectives require internal representations to support other learning goals besides the primary task of value learning. Auxiliary objectives thus regularize internal representations to preserve information that may be relevant for learning. They are thought to address challenges that may arise in sparse reward environments, such as representation collapse and value overfitting (Lyle et al., 2021). Many auxiliary objectives used in machine learning are predictive in flavor. Prior work has found success in defining objectives to predict reward (Jaderberg et al., 2016; Shelhamer et al., 2016) or to predict future states (Shelhamer et al., 2016; Oord et al., 2018; Wayne et al., 2018) from history. Predictive objectives may be useful for additional functions as well. Intrinsic rewards based on the agent's ability to predict the next state can be used to guide curiosity-driven exploration (Pathak et al., 2017; Tao et al., 2020). These objectives may also aid with transfer learning (Walker et al., 2023), by learning representations that capture features that generalize across diverse domains. The incorporation of auxiliary objectives has greatly enhanced the efficiency and robustness of deep RL models in machine learning applications.

In neuroscience, much theoretical work has sought to characterize brain regions by the computational objective they may be responsible for. Hippocampus in particular has been suggested to learn predictions of an animal's upcoming experience. This has been formalized as learning a transition model similar to model-based reinforcement learning (Fang et al., 2022) to learning long-horizon predictions as in the successor representation (Gershman et al., 2012; Stachenfeld et al., 2017). Separately, the striatum has long been suggested to support model-free (MF) reinforcement learning like actor-critic models (Joel et al., 2002), with more recent work connecting these hypotheses to deep RL settings (Dabney et al., 2021; Lindsey & Litwin-Kumar, 2022).

Less work has been done to understand how the computational objectives of multiple brain regions interact, although this has been suggested as a framework for neuroscience (Marblestone et al., 2016; Yamins & DiCarlo, 2016; Botvinick et al., 2020). Prior work has used multi-region recurrent neural networks (Pinto et al., 2019; Andalman et al., 2019; Kleinman et al., 2021) or switching nonlinear

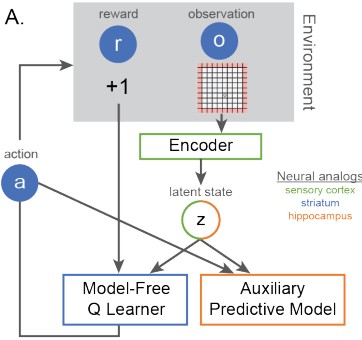

Figure 1: A deep RL framework to model multi-region computation. **A.** In the deep RL model we use, reward is provided as a scalar input $r$. Observations $o$ are 2D visual inputs fed into an encoder (green) that learns low-dimensional state space representations $z$. The encoder is a convolutional neural network. Representations $z$ are used to learn Q values via a MLP (blue); these Q values are used to select actions $a$. A predictive auxiliary objective (orange) is enforced by a separate MLP learning predictions from $z$.

dynamical systems (Semedo et al., 2014; Glaser et al., 2020; Karniol-Tambour et al., 2022) to model the interactions of different regions. However, much of this work focuses more on fitting recorded neural activity than taking a normative perspective on brain function. A growing body of work considers modular and multi-objective approaches to building integrative models of brain function. One approach has been to construct multi-region models by combining modules performing independent computations and comparing representations in these models to neural activity (Frank & Claus, 2006; O'Reilly & Frank, 2006; Geerts et al., 2020; Russo et al., 2020; Liu et al., 2023; Jensen et al., 2023). On the behavioral end, there has also been prior work discussing how the addition of biologically-realistic regularizers or auxiliary objectives can result in performance more consistent with humans (Kumar et al., 2022; Binz & Schulz, 2022; Jensen et al., 2023).

Our work differs in that the entire system consists of a neural network that is trained end-to-end, allowing us the opportunity to specifically study the effects on representation learning. In this paper, we show how deep RL networks can be a testbed for studying representational changes and serve as a multi-region model for neuroscience.

## 3 EXPERIMENTAL METHODS

**Network architecture** We implement a double deep Q-learning network (Van Hasselt et al., 2016) with a predictive auxiliary objective, similar to François-Lavet et al. (2019) (Fig 1A). A deep convolutional neural network $E$ encodes observation $o_t$ at time $t$ into a latent state $z_t$ ($o_t$ will be a 2D image depicting the agent state in a tabular grid world). The state $z_t$ is used by two network heads: a Q-learning network $Q(z, a)$ that will be used to select action $a_t$ and a prediction network $T(z, a)$ that predicts future latent states. Both $Q$ and $T$ are multi-layer perceptrons with one hidden layer.

**Network training procedure** The agent is trained on transitions $(o_t, a_t, o_{t+1}, a_{t+1})$ sampled from a random replay buffer. We will also let $o_i$ and $o_j$ denote any two observations randomly sampled from the replay buffer that may not have occurred in sequence. The weights of $E$, $Q$, $T$ are trained end-to-end to minimize the standard double Q-learning temporal difference loss function $\mathcal{L}_Q$ Van Hasselt et al. (2016) and a predictive auxiliary loss $\mathcal{L}_{pred}$. The predictive auxiliary loss is similar to that of contrastive predictive coding (Oord et al., 2018). That is, $\mathcal{L}_{pred} = \mathcal{L}_+ + \mathcal{L}_-$ where $\mathcal{L}_+$ is a positive sampling loss and $\mathcal{L}_-$ is a negative sampling loss. The positive sample loss is defined as $\mathcal{L}_+ = ||\tau(z_t, a_t) - z_{t+1} - \gamma\tau(z_{t+1}, a_{t+1})||^2$, where $z_t = E(o_t)$ and $\tau(z_t, a_t) = z_t + T(z_t, a_t)$. That is, in the $\gamma = 0$ case, the network $T$ is learning the difference between current and future latent states such that $\tau(z_t, a_t) = z_t + T(z_t, a_t) \approx z_{t+1}$. This encourages the learned representations $z$ to be structured to be structured so as to be consistent with predictable transitions (François-Lavet et al., 2019). Additionally, $\gamma$ modulates the predictive horizon.

The negative sample loss is defined as $\mathcal{L}_- = -\exp||z_i - z_j||$. We emphasize that $z_i$ and $z_j$ are randomly sampled from the buffer and thus may represent states that are spatially far from another. This loss drives temporally distant observations to be represented differently, thereby preventing the trivial solution from being learned (mapping all latent states to a single point). The use of two contrasting terms ($\mathcal{L}_-$ and $\mathcal{L}_+$) is not just useful for optimization reasons– it also mirrors the hypothesized pattern separation and pattern completion within the hippocampus (O'Reilly & McClelland, 1994; Schapiro et al., 2017). However, we note that negative sampling elements are not always needed to support self-predictive learning if certain conditions are satisfied (Tang et al., 2023). Except where indicated, the agent learns off-policy via a random policy during learning, only using its policy during test time. The weights over loss terms $\mathcal{L}_Q$, $\mathcal{L}_+$, $\mathcal{L}_-$ are chosen through a small grid search over the final episode score.

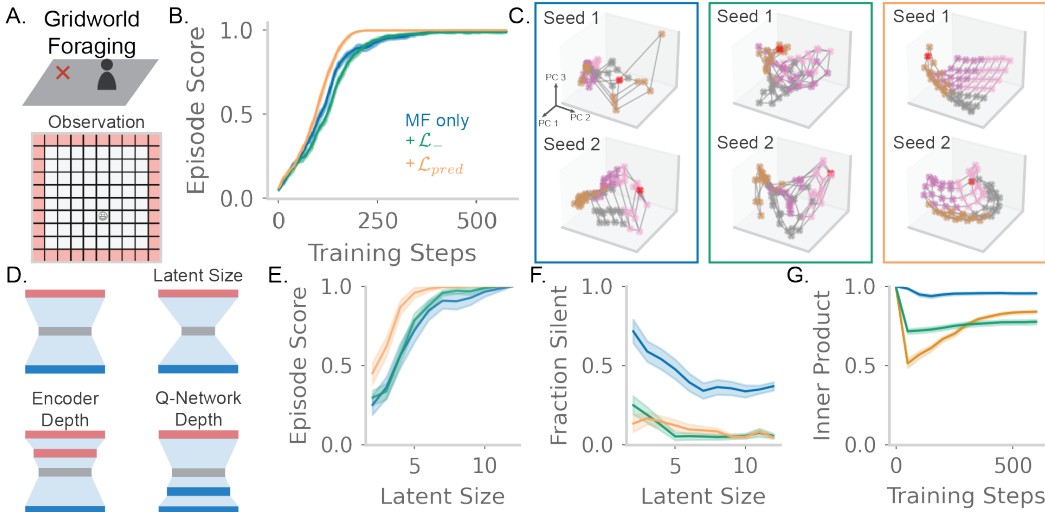

Figure 2: Gridworld performance with predictive auxiliary tasks. **A.** The model is tested on grid-world task in a 8x8 arena. The agent must navigate to a hidden reward given random initial starting locations. **B.** Average episode score across training steps for models without auxiliary losses (blue), with only the negative sampling loss $\mathcal{L}_-$ (green), and with the full predictive loss $\mathcal{L}_{pred}$ (orange). The maximum score is 1 and $|z| = 10$ (i.e. $z$ contains 10 units). In each step, the network is trained on one batch of replayed transitions (batch size is 64). All error bars are standard error mean over 45 random seeds. **C.** 3D PCA representations of latent states $z$ for the models in (B) (two random seeds). The latent states are colored by the quadrant of the arena they lie in. The quadrants (in order) are purple, pink, gray, brown. The goal location state is colored red. Gray lines represent the true connectivity between states. **D.** Diagram of the encoder network (red), learned latent state (gray), and value-learning network (blue). We vary $|z|$ (see E, F), as well as the encoder/decoder depths (Appendix A.3AB). **E.** Average episode score at the end of learning (600 training steps) across $|z|$. **F.** Fraction of units in $z$ that are silent during the task, across $|z|$. **G.** Cosine similarity of two randomly sampled states throughout learning, $|z| = 10$.

**Experimental comparisons and modifications** We will treat the encoder network as a sensory cortex analog, the Q-learning network as a striatum analog, and the prediction network as a hippocampus analog (Fig A.1AB). In our analyses, we vary several parameters of interest. We vary the size of $z$ to test the effects of the information bottleneck of the encoder. We will also modulate the strength of $\gamma$ in the auxiliary loss to test the effects of different timescales of prediction. Finally, we also test how the depths of the decoder and encoder networks affect learning.

## 4 RESULTS

### 4.1 PREDICTIVE OBJECTIVES HELP PREVENT REPRESENTATIONAL COLLAPSE.

We first want to understand the effect predictive auxiliary objectives have on a learning system. We test the RL model in a simple gridworld foraging task, where an agent must navigate to a hidden reward from any point in a 2D arena. The observation received by the agent is a 2D image depicting a birds-eye view of the agent's location. Further details and examples are provided in Figure A.2 A-D. We compare a model without auxiliary objectives (MF-only) to models with the negative sampling objective $\mathcal{L}_-$ only and with the full predictive objective $\mathcal{L}_{pred}$. Here, the predictive model is trained with one-step prediction ($\gamma = 0$).

Given sufficient capacity in the encoder, decoder, and latent layer $z$, all models learn the foraging task (Fig 2B). However, the model with prediction reaches maximum performance with fewer training steps than both the negative-sampling model and the MF-only agent (Fig 2B). Additionally, the latent representation in the predictive model appears to capture the global structure of the environment better than the other two models (Fig 2C). The model without any auxiliary tasks tends to expand the representation space around rewarding states, while the model with negative sampling (Fig 2C) evenly spaces apart state representations without regard for environment structure.

We next tested how the effects of auxiliary tasks change with the size of the model components (Fig 2D). We first varied the size of $z$, and thus the representational capacity of the encoder. We find that, although all models can perform well given a large enough latent dimension $|z|$, supplying the model with a predictive auxiliary objective allows the model to learn the task even with a smaller bottleneck (Fig 2E). This benefit is not conveyed by the negative sampling loss alone, suggesting that learning the environment structure confers its own unique benefit (Fig 2E). We find similar results by varying the encoder network depth and the decoder network depth (Fig A.3AB), showing that the benefits of predictive auxiliary objectives are more salient in resource-limited cases.

This difference may be because representational collapse is a greater danger in lower-dimensional settings. To test this, we measure how many units in the output of the encoder are involved in supporting the state representation. We find that a greater proportion of units are completely silent in the MF-only encoder (Fig 2F), suggesting a less distributed representation. To more directly test for collapse, we measure how the cosine similarity between state representations change across learning. Although all models start with highly similar states, the models with auxiliary losses separate state representations across training more than the MF-only model does (Fig 2G).

Finally, we test more complex versions of this gridworld task to see how performance is affected. We find consistent results in a CIFAR version of this task (Fig A.2D), where models equipped with a predictive auxiliary objective outperform the other two models we tested (Fig A.3C). We also test a version of gridworld where the environment is less predictable– that is, transitions are no longer determinstic. We find that, as the probability of stochastic transitions increse, the benefit of predictive auxiliary objectives vanish (Fig A.3D).

## 4.2 LONG-HORIZON PREDICTIVE AUXILIARY TASKS ARE MORE EFFECTIVE AT SUPPORTING REPRESENTATIONAL TRANSFER THAN SHORT-HORIZON PREDICTIVE TASKS.

Thus far, we have tested the predictive auxiliary objective with one-step prediction. However, long horizon predictions are often used as auxiliary objectives (Oord et al., 2018; Hansen et al., 2019), and many neural systems, including hippocampus, have been hypothesized to perform long-horizon predictions (Brunec & Momennejad, 2022; Lee et al., 2021). We next sought to understand under what conditions longer horizons of prediction in auxiliary objectives would be useful. In particular, we were interested in exploring how well learned representations could transfer to new tasks. We hypothesize that long-horizon predictions (larger $\gamma$ in $\mathcal{L}_+$) can better capture global environment structure and thus learn representations that transfer better to tasks in similar environments.

We first test representation transfer to new reward locations in gridworld (Fig 3A). After the agent learns an initial goal location in task A, we freeze the encoder, move the goal to a new state, and fine-tune the value network for task B. This allows us to test how well the learned representation structure can support new value functions. We test models with $\mathcal{L}_{pred}$ loss and $\gamma \in \{0.0, 0.25, 0.5, 0.8\}$. We find that, although all models learn task A quickly, models with larger $\gamma$ learn task B more efficiently (Fig 3B). We test how this effect scales with latent sizes. Just having a predictive horizon longer than one timestep appears sufficient to improve learning efficiency, with the effect stronger at larger latent sizes. (Fig 3C). The selective benefit of longer time horizons for transfer may explain the observation that regions of hippocampus with larger spatial scales appear to be preferentially active in novel environments (Fredes et al., 2021; Köhler et al., 2002; Poppenk et al., 2010).

We hypothesize that the difference in efficient transfer performance across the models may result from learning a latent structure that better reflects global structure. Long-horizon prediction may be better at smoothing across experience over many timesteps, thus capturing global environment structure better than short-horizon prediction and providing a larger benefit when latent representations are higher dimensional. Indeed, models with smaller $\gamma$ values tend to learn more curved maps that preserve local, but not global, structure (Fig 3D). To quantify this effect, we measured the inner product between the states representing the corners of the environment. These are states that are maximally far from each other, and as such, representations that capture the environment structure accurately should separate these states from each other. We see that, across learning, models with larger $\gamma$ learn to separate corner states better (Fig 3E).

Predictive auxiliary objectives can also be disadvantageous under certain regimes. Predictive objectives shape latent representations to reflect transition structure. However, these learned representations might not generalize well to new tasks where the transition structure or the policy changes. We

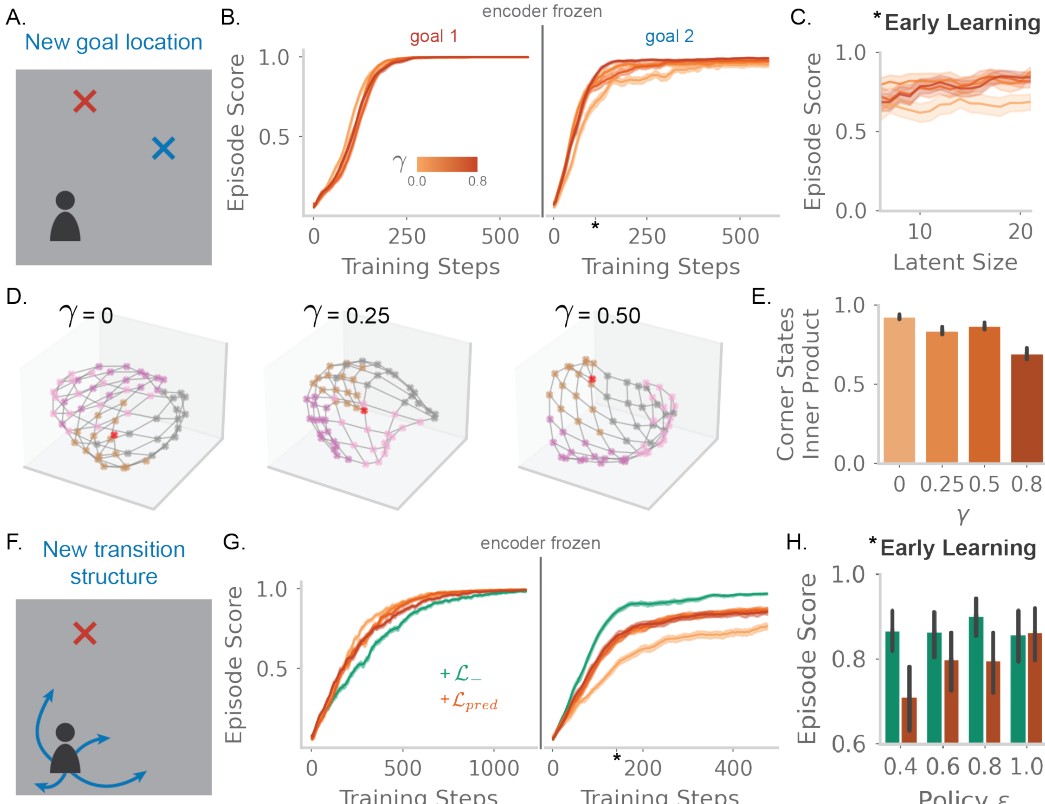

Figure 3: Effects of predictive auxiliary objectives across transfer learning scenarios. **A.** We test goal transfer by moving the goal location to a new state in task B. After training on task A, encoder weights are frozen and the value function is fine-tuned on task B. **B.** Average episode score across task A, then task B. All models shown use the predictive auxiliary loss, with the shade of each line corresponding to the magnitude of $\gamma$ in $\mathcal{L}_{pred}$ ($\gamma \in \{0.0, 0.25, 0.5, 0.8\}$, $|z| = 17$). **C.** The episode score after 100 training steps for each of the models in (B), as $|z|$ is increased. All models achieve maximum performance in task A. 30 random seeds are run for each latent size. **D.** 3D PCA plots, for three models ($\gamma = 0.0, 0.25, 0.5$) with the same random seed. **E.** Pairwise cosine similarity values between the corner states of the arena for the model shown in (B). **F.** We test transition transfer by shuffling the connectivity between all states in task B. Freezing and fine-tuning are the same as in (A). **G.** Average episode score across task A, then task B. Here, $|z| = 17$ and $\epsilon = 0.4$-greedy policy during learning. In green is the model with only $\mathcal{L}_-$ as an auxiliary loss. **H.** Episode score after 150 training steps for the model with only $\mathcal{L}_-$ (green) versus the model with $\mathcal{L}_{pred}$ for $\gamma = 0.8$. On the x-axis, the policy $\epsilon$ used during training is varied, with $\epsilon = 1.0$ corresponding to a fully random policy ($|z| = 17$, all models achieve maximum performance on task A).

test this in a different transfer task, where reward location remains the same in task B, but the environment transition structure is scrambled (Fig 3F). Additionally, to test for effects of policy change across task A and B, we vary the portion of random actions taken in our $\epsilon$-greedy agent. Under this new transfer task with $\epsilon = 0.4$, we find performance in task B decreases for models with the predictive objective compared to a model with just the negative sampling loss. (Fig. 3G).

Indeed, as $\epsilon$ gets smaller and the agent learns more from biased on-policy transition statistics, transfer performance on task B accordingly suffers (Fig 3G,H). All models with predictive objectives do not perform as well in task B as a model with only negative sampling loss (Fig 3G,H).

### 4.3 EFFECTS OF VALUE LEARNING AND HISTORY-DEPENDENCE IN PREDICTION NETWORK RESEMBLE HIPPOCAMPAL ACTIVITY.

We next ask how well representations developed in the network can model representations found in neural activity. The output of our $T$ network serves as an analog to the hippocampus, a region

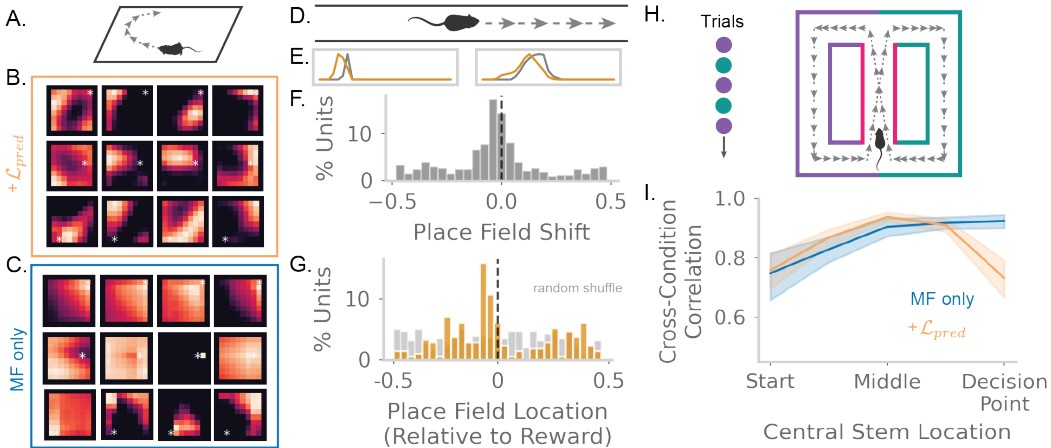

Figure 4: Representational changes in the predictive model are similar to those observed in the hippocampus. **A.** 2D foraging experiments are simulated as in the gridworld task from Fig 1-2. **B.** 2D receptive fields from top four $T$ units (columns) sorted by spatial information score (Skaggs et al., 1992). Three random seeds are shown (rows). The model uses $\mathcal{L}_{pred}$ and $|z| = 10$. White asterisk depicts reward. **C.** As in (B), but the model has no auxiliary objectives. **D.** Circular track experiments are simulated in a circular gridworld with 28 states. Reward is in a random state for each seed and the agent is rewarded for running clockwise to the reward. **E.** Receptive fields of two example units in the $T$ network before (gray) and after (orange) learning. **F.** Histogram over the shift in receptive field peaks for units in $T$ over 15 random seeds, where $|z| = 24$. Positive values indicate shifts forward, and vice-versa for negative values. Black dotted line at 0. Median of the histogram is $-0.034$. **G.** Histogram over the location of receptive field peaks for units in (F), with location centered around the reward site. Random shuffle (gray) control was made by randomly shuffling the weights of the $T$ network. Black dotted line at 0. The model median is $-0.06$, while the random shuffle median is $-0.02$. **H.** We simulate a 5x5 alternating-T maze (see Appendix); center corridor in pink. **I.** Cosine similarity of $T$ population vector responses in the center corridor under left-turn versus right-turn conditions. X-axis depicts location in the center corridor. Data is from 20 random seeds. Shown is the model without auxiliary objectives (blue) and the model with $\mathcal{L}_{pred}$ (orange). $T$ is randomly initialized for the model without an auxiliary objective.

implicated in self-predictive learning. We first test whether the $T$ network activity can capture a classic result in the hippocampal literature: formation of spatially local activity patterns, called place fields. We plot the spatial firing fields of individual $T$ units in our model trained on gridworld, and find 2D place fields as expected (Fig 4B). We also find that the prevalence of these place fields is greatly reduced in models without predictive auxiliary tasks (Fig 4C).

Hippocampal place fields also undergo experience-dependent changes. We test for these effects in our model through 1D circular track experiments (Fig 4D). We find that place fields developed on the 1D track will skew and shift backwards from the movement of the animal (Fig 4E,F). This is consistent with phenomena in rodent hippocampal data that have been attributed to predictive learning (Mehta et al., 2000). We also find that the number of place fields across the linear track is more abundant close to the reward site (Fig 4G), another widely observed phenomena that is considered to be a result of reward learning in hippocampus. Our results suggest that value learning in shared representations with other systems can result in reward-related effects in the hippocampus.

Finally, we test a more complex form of experience-dependency in neural activity by simulating an alternating T-maze task. In this task, animals alternate between two trial types: one where they run down a center corridor and turn left for reward, and another where they run down the same center corridor but turn right for reward. In these tasks, neural activity has been observed to "split" – neurons fire differently in the center corridor across trial types despite the spatial details of the corridor remaining the same (Duvelle et al., 2023). Interestingly, the degree of splitting is greatest in the beginning of the corridor and also high at the end of the corridor, splitting least in the middle of the corridor (Duvelle et al., 2023). To enable the agent to perform this task, which requires remembering

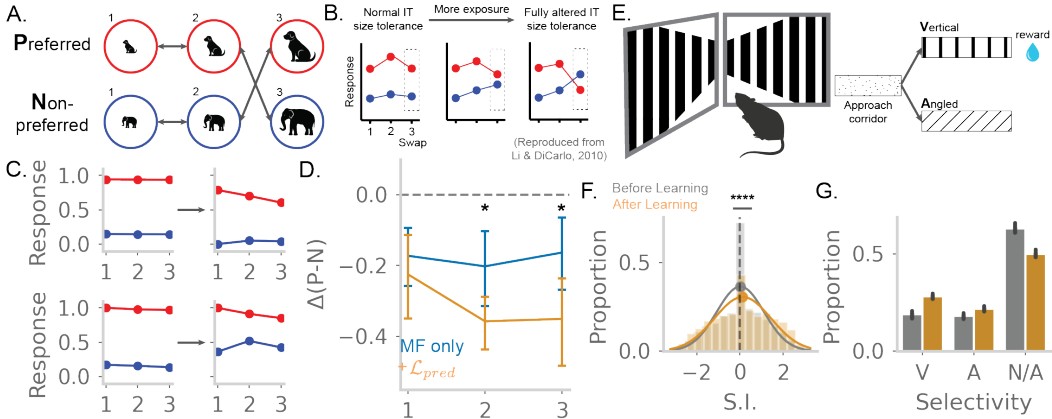

Figure 5: Representational changes in the encoder model resemble recordings from visual cortex. **A.** Example sequence structure in the preference swap task of Li & DiCarlo (2008; 2010), images numbered by seqeunce location. **B.** Example changes in IT neuron response to preferred images (red) and non-preferred images (blue) across exposure to new image transitions. **C.** Responses of two example units from the model with $\mathcal{L}_{pred}$. Arrows indicate response profile before and after experiencing swapped transitions. Red indicates the response to $P1, P2, P3$ states that were selected from the gridworld environment, while blue indicates the response to $N1, N2, N3$ states selected from the environment. **D.** Change in response difference between $(P1, N1)$, $(P2, N2)$, and $(P3, N3)$ over 10 units. Each unit is a separate transition swap experiment. Shown is the model without any auxiliary objectives (blue) and the model with $\mathcal{L}_{pred}$ (orange). Asterisks indicate significance from a t-test comparing the means from both models. We additionally note that the means of both models are significantly different from 0. **E.** Linear track VR experiment used in Poort et al. (2015). Vertical stripe corridors were rewarded but angled corridors were not. Animals experienced either condition at random following an approach corridor. **F.** Selectivity across the population before learning (gray) and after learning (orange). Selectivity was calculated as in Poort et al. (2015), with negative and positive values corresponding to angled and vertical corridor preference, respectively. Asterisks indicate significance from one-tailed t-test ($t = -12.43$, $p = 9 \times 10e-36$) **G.** Selectivity of individual units before and after learning for vertical condition (V), angled condition (A), or neither (N/A). Units are pooled across 15 experiments.

the previous trial type, we introduce a memory component to the agent so that a temporally graded trace of previous observations are made available. That is, the input into the encoder at time $t$ is $o_t + \alpha o_{t-1} + \alpha^2 o_{t-2} + \ldots$ for some $\alpha < 1$. This decaying sum of recent observations captures information about the recent past in a simple way, and is inspired by representations hypothesized by temporal context model (Howard & Kahana, 2002). We measure cosine similarity between population activity in the left turn condition and the right turn condition. Lower similarity corresponds to greater splitting. The representations in both a MF-only model and the model with the predictive objective show increased splitting in the beginning of the corridor due to the memory component (Fig 4F). However, only the model with the predictive objective shows increased splitting at the end of the corridor (Fig 4F). This shows that the pattern of splitting seen in data can be captured by a model using both memory and prediction.

We also test the effects of recurrency in the model by simulating a partially observable version of the alternating-T maze (Fig A.5C). To solve this version of the task, recurrency must be used to infer the current latent state with the model's previous latent state. We find consistent results where the inclusion of a predictive auxiliary objective greatly improves the model's ability to learn the task (Fig A.5DE) and where only the model with the predictive objective shows a splitting pattern consistent with data (Fig A.5F).

### 4.4 EFFECTS OF VALUE LEARNING AND TRANSITION LEARNING IN THE ENCODER NETWORK RESEMBLE ACTIVITY IN VISUAL CORTEX.

As another example of representational effects arising from mutually interacting regions, we compare the activity of our encoder network to experimental results in sensory cortices. Neurons in

visual cortex (even those in primary regions) have been observed to change their tuning as a result of learning Poort et al. (2015); Li & DiCarlo (2008; 2010); Wilmes & Clopath (2019); Pakan et al. (2018). Our model provides a simple system to look for such effects.

First, we test for effects of prediction and temporal statistics that have been seen in visual cortex. Specifically, Li & DiCarlo (2008) found that object selectivity in macaque IT neurons could be altered by exposing animals to sequences of images where preferred stimuli and non-preferred stimuli became linked (Fig 5A). The images in the preferred and non-preferred that are linked together are referred to as the "swap position" within a sequence (Fig 5A). An analogous experiment can be run in our gridworld task from Fig 2. We first identify spatially contiguous preferred and non-preferred states of neurons in the encoder network. We then expose the model to sequences where preferred states and non-preferred states became connected at arbitrarily chosen swap positions (Fig 5B). We find neurons in the output of the encoder that, after exposure, decrease their firing rate for the preferred stimulus at the swap location and increase their firing rate for the non-preferred stimulus at the swap position (Fig 5C). This is consistent with observations in data as well (Li & DiCarlo, 2008; 2010). We quantify this change in firing rate at different sequence locations. We find a similar trend as in data, where tuning for stimuli closer to the swap position is increasingly altered away from the original preferences (Fig 5D). Importantly, this effect is not present without the predictive auxiliary objective, similar to lesion studies carried out in Finnie et al. (2021).

The downstream Q-learning objective also have an effect on representations in the encoder. We simulate value learning effects in visual cortical activity through linear track experiments used in Poort et al. (2015) (Fig 5E). In this experiment, authors found that V1 neurons in mice increased selectivity for visual cues in the environment after learning the task. Furthermore, the authors noted a slight selectivity increase for more rewarding cues (vertical gratings) compared to nonrewarding cues (angled gratings). We find a similar effect in units in early layers of the encoder network: a small, but statistically significant increase in proportion of units encoding the rewarded stimulus (Fig 5F). As in Poort et al. (2015), selectivity increases across learning, but with a greater preference for the vertical grating environment (Fig 5G).

## 5 CONCLUSION

In this work, we explore the representational effects induced by predictive auxiliary objectives. We show how such objectives are useful in resource-limited settings and in certain transfer learning settings. We also investigate how prediction and predictive horizons affect learned representation structure. Furthermore, we describe how such deep RL models can function as a multi-region model for neuroscience. We show how representation learning in the prediction model recapitulates experimental observations made in the hippocampus. We make similar connections between representation learning in the encoder model and learning in visual cortex.

Our results point to a new perspective on the role of the hippocampus in learning. That is, a predictive system like the hippocampus can be useful for learning without being used to generate sequences or support planning. Learning predictions is sufficient to induce useful structure into representations used by other regions. This view also connects to trends seen in machine learning literature. In deep RL, predictive models need not be used for forward planning (Hamrick et al., 2020) to be useful for representation learning. Additionally, the contrastive prediction objective used in this work is drawn from machine learning literature but bears interesting similarities to classic descriptions of hippocampal computation. CA3 and CA1 in the hippocampus have been implicated in predictive learning similar to the positive sampling loss. Meanwhile, the dentate gyrus in the hippocampus has been proposed to perform pattern separation similar to the contrastive negative sampling loss.

Our results are limited in the complexity of tasks and the diversity of auxiliary objectives tested. Future work can improve on current understanding by more systematically comparing effects across objectives over more complex tasks. We also did not examine representations in the value learning network, which is ripe for comparison with striatum data. Future work can also explore the effects of recurrence across modules, which can be both functionally useful and more biologically realistic.

Overall, this work points to the utility of a modeling approach that considers the effect of multiple objectives in a deep learning system. The deep network setting reveals new aspects of neuroscience modeling that are less apparent in tabular settings or in simpler models.

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
