## A  APPENDIX

### A.1  ALTERNATING-T MAZE SIMULATION

The maze is $5\times5$. That is, the agent enters the center stem at $(x = 2, y = 0)$ and reaches the decision point at $(x = 2, y = 4)$. The agent is incentivized to follow a figure-8 path via invisible barriers and the presence of reward at $(0, 4)$, $(2, 4)$, and $(4, 4)$. The model is simulated with a 6-frame memory trace with weight decay of $0.9$.

### A.2  GRIDSEARCH FOR LEARNING RATES

Weights below are formatted as [Q Loss, $\mathcal{L}_-$, $\mathcal{L}_+$]

- MF Only: [[1e−5, 0, 0], [1e−4, 0, 0], [1e−3, 0, 0]]
- MF + Negative Sampling: [[1e−4, 1e−6, 0], [1e−4, 1e−5, 0], [1e−4, 1e−4, 0], [1e−4, 1e−3, 0], [1e−4, 1e−2, 0]]
- MF + Positive Sampling, $\gamma = 0$: [[1e−4, 1e−6, 1e−6], [1e−4, 1e−5, 1e−6], [1e−4, 1e−4, 1e−6], [1e−4, 1e−3, 1e−6]]
- MF + Positive Sampling, $\gamma = 0.25$: [[1e−4, 1e−6, 1e−6], [1e−4, 1e−5, 1e−6], [1e−4, 1e−4, 1e−6], [1e−4, 1e−3, 1e−6], [1e−4, 1e−6, 1e−7], [1e−4, 1e−5, 1e−7], [1e−4, 1e−4, 1e−7], [1e−4, 1e−3, 1e−7]]
- MF + Positive Sampling, $\gamma = 0.5$: [[1e−4, 1e−6, 1e−6], [1e−4, 1e−5, 1e−6], [1e−4, 1e−4, 1e−6], [1e−4, 1e−3, 1e−6], [1e−4, 1e−6, 1e−7], [1e−4, 1e−5, 1e−7], [1e−4, 1e−4, 1e−7], [1e−4, 1e−3, 1e−7]]
- MF + Positive Sampling, $\gamma = 0.8$: [[1e−4, 1e−6, 1e−7], [1e−4, 1e−5, 1e−7], [1e−4, 1e−4, 1e−7], [1e−4, 1e−3, 1e−7], [1e−4, 1e−6, 1e−8], [1e−4, 1e−5, 1e−7], [1e−4, 1e−4, 1e−8], [1e−4, 1e−3, 1e−8]]

### A.3  PARAMETERS FOR BASE NETWORK

| Module | Layer | Activation |
|---|---|---|
| Encoder Network | Conv2D: 16 channels, 2x2 kernel size | ReLU |
| | Conv2D: 32 channels, 2x2 kernel size | ReLU |
| | MaxPool2D, 2x2 kernel size | |
| | Fully Connected: output size 32 | ReLU |
| | Fully Connected: output size $|z|$ | ReLU |
| Q Network | Fully Connected: output size 16 | ReLU |
| | Fully Connected: output size 1 | |
| T Network | Fully Connected: output size 16 | ReLU |
| | Fully Connected: output size $|z|$ | |

| Model | Q Loss | $L_-$ | $L_+$ |
|---|---|---|---|
| MF Only | 1e−4 | 0 | 0 |
| MF + Negative Sampling | 1e−4 | 1e−4 | 0 |
| MF + Negative & Positive Sampling, $\gamma = 0$ | 1e−4 | 1e−5 | 1e−6 |
| MF + Negative & Positive Sampling, $\gamma = 0.25$ | 1e−4 | 1e−4 | 1e−6 |
| MF + Negative & Positive Sampling, $\gamma = 0.5$ | 1e−4 | 1e−4 | 1e−7 |
| MF + Negative & Positive Sampling, $\gamma = 0.8$ | 1e−4 | 1e−4 | 1e−8 |

Table 1: Learning Rates

## A.4 PARAMETERS FOR NETWORK WITH DEEPER ENCODER

| Module | Layer | Activation |
|---|---|---|
| Encoder Network | Conv2D: 16 channels, 2x2 kernel size | ReLU |
| | Conv2D: 48 channels, 2x2 kernel size | ReLU |
| | MaxPool2D, 2x2 kernel size | |
| | Fully Connected: output size 48 | ReLU |
| | Fully Connected: output size 32 | ReLU |
| | Fully Connected: output size $|z|$ | ReLU |
| Q Network | Fully Connected: output size 16 | ReLU |
| | Fully Connected: output size 1 | |
| T Network | Fully Connected: output size 16 | ReLU |
| | Fully Connected: output size $|z|$ | |

| Model | Q Loss | $L_-$ | $L_+$ |
|---|---|---|---|
| MF Only | 1e−4 | 0 | 0 |
| MF + Negative Sampling | 1e−4 | 1e−4 | 0 |
| MF + Negative & Positive Sampling | 1e−4 | 1e−5 | 1e−6 |

Table 2: Learning Rates

## A.5 PARAMETERS FOR NETWORK WITH DEEPER Q NETWORK

| Module | Layer | Activation |
|---|---|---|
| Encoder Network | Conv2D: 16 channels, 2x2 kernel size | ReLU |
| | Conv2D: 32 channels, 2x2 kernel size | ReLU |
| | MaxPool2D, 2x2 kernel size | |
| | Fully Connected: output size 32 | ReLU |
| | Fully Connected: output size $|z|$ | ReLU |
| Q Network | Fully Connected: output size 32 | ReLU |
| | Fully Connected: output size 16 | ReLU |
| | Fully Connected: output size 1 | |
| T Network | Fully Connected: output size 16 | ReLU |
| | Fully Connected: output size $|z|$ | |

| Model | Q Loss | $L_-$ | $L_+$ |
|---|---|---|---|
| MF Only | 1e−4 | 0 | 0 |
| MF + Negative Sampling | 1e−4 | 1e−2 | 0 |
| MF + Negative & Positive Sampling, $\gamma = 0$ | 1e−4 | 1e−5 | 1e−6 |
| MF + Negative & Positive Sampling, $\gamma = 0.25$ | 1e−4 | 1e−4 | 1e−6 |
| MF + Negative & Positive Sampling, $\gamma = 0.5$ | 1e−4 | 1e−4 | 1e−6 |
| MF + Negative & Positive Sampling, $\gamma = 0.8$ | 1e−4 | 1e−4 | 1e−8 |

Table 3: Learning Rates

A.6   SUPPLEMENTARY FIGURES

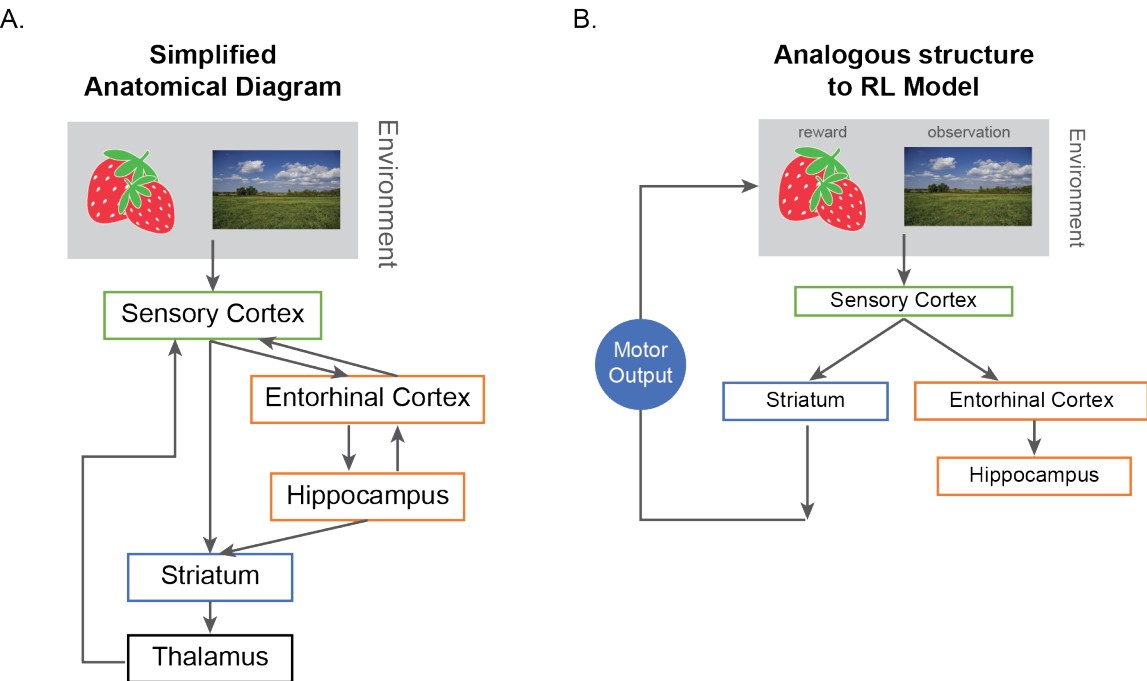

Figure A.1: **A.** Simplified diagram of brain regions of interest. Although not exhaustive, this diagram shows relevant connections between the regions of interest (Canto et al., 2008; Goodroe et al., 2018; Morgenstern et al., 2022). **B.** Further simplified diagram from (A). This version highlights systems in the brain that are analogous to the encoder, model-free value learning system, and predictive auxiliary task described in 1(A).

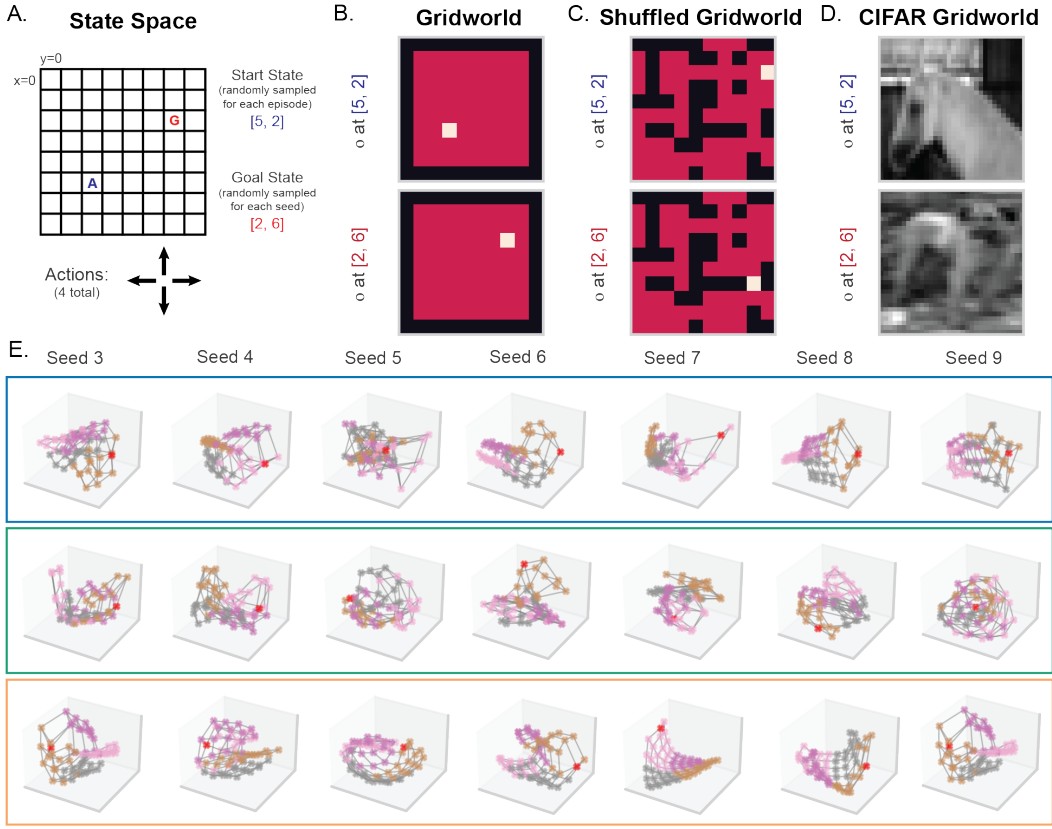

Figure A.2: **A.** We use a $8 \times 8$ gridworld environment. Shown is an example where the agent starts at location $[5, 2]$ and the goal state is at $[2, 6]$. Four actions are possible: left, right, top, and bottom. **B.** 2D visual observations provided to the agent at the start and goal state shown in (A). Note that only agent, and not goal, location is visible. These are the observations used to visualize latents in PCA plots. **C.** A version of gridworld where the visual observations are as in (B), but randomly shuffled. These are the observations used to compare performances across models. **D.** A version of gridworld where the observation at each state is a randomly selected CIFAR-10 image. **E.** As in Figure 2C, but for seven more additional seeds.

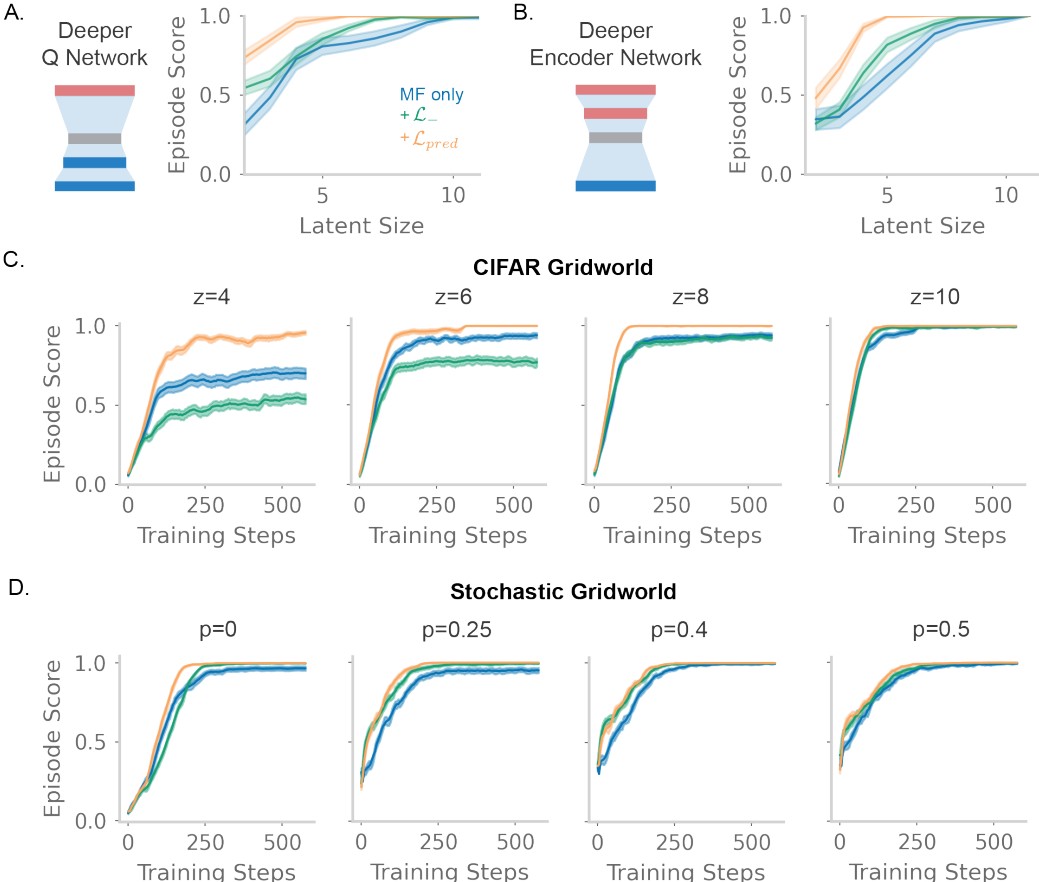

Figure A.3: **A.** As in Figure 2E, but for the model with a deeper Q network. **B.** As in Figure 2E, but for the model with a deeper encoder network. **C.** Models as in Figure 2B, but for CIFAR gridworld environment across latent sizes $z = \{4, 6, 8, 10\}$. **D.** Models as in Figure 2B, but in a shuffled gridworld environment with stochastic transitions. That is, if the agent selects action $a$ at some timestep, with probability $p$ the environment transition instead randomly follows the transition of one of the three other actions that is not $a$. Here, examples are shown for $p = \{0., 0.25, 0.4, 0.5\}$.

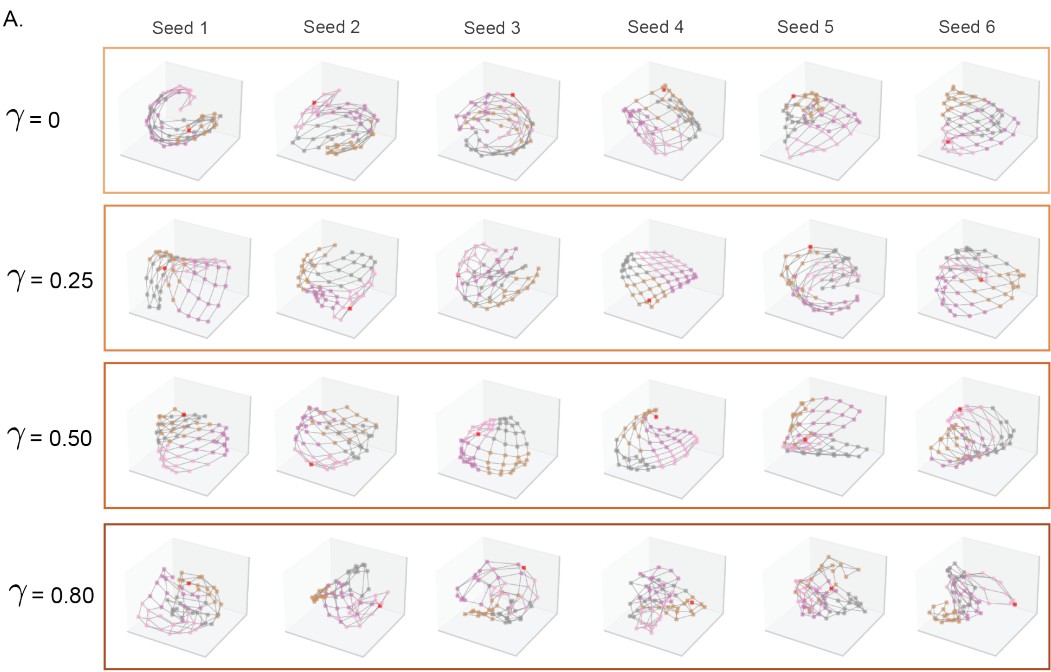

Figure A.4: **A.** As in Figure 3D, but for six additional seeds, and including $\gamma = 0.8$.

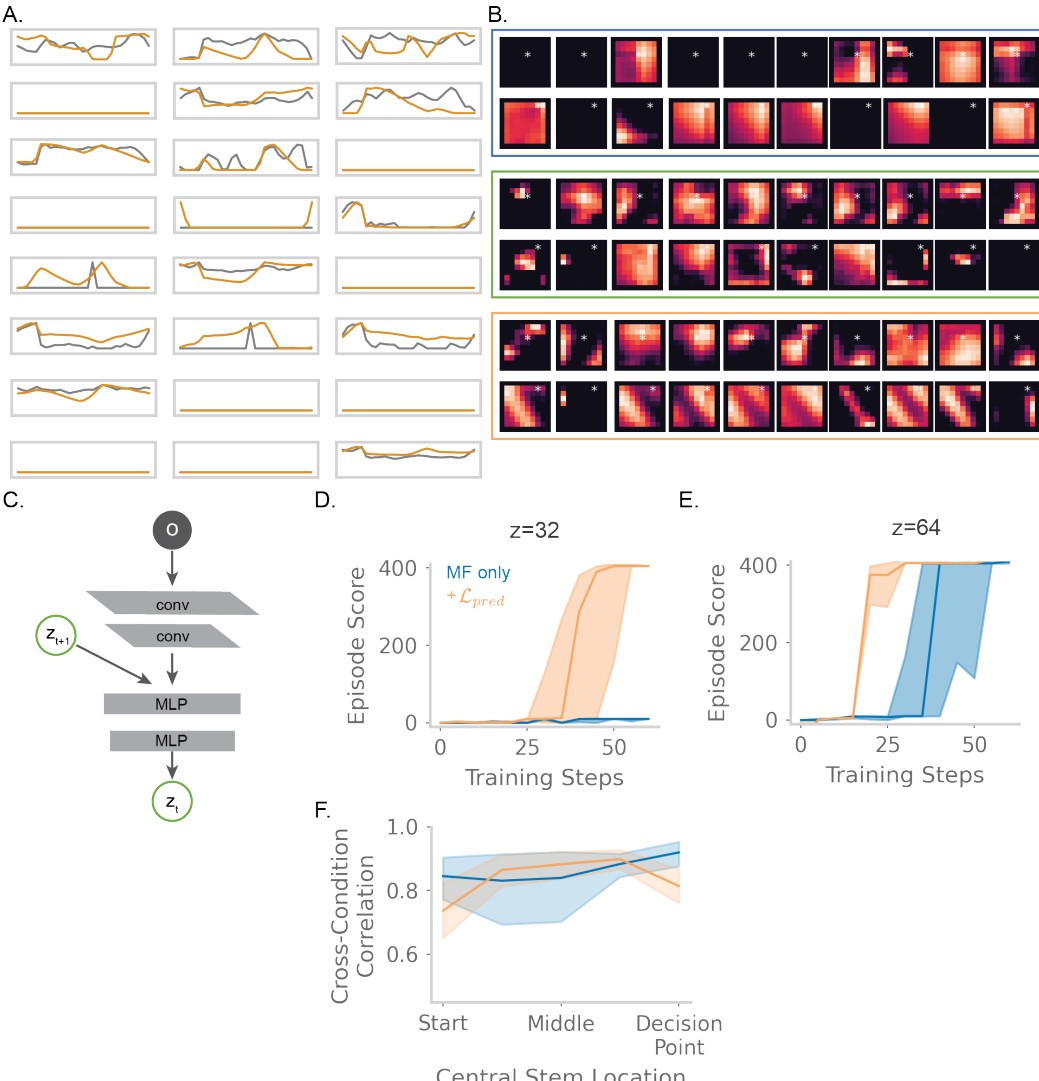

Figure A.5: **A.** As in Figure 4B, but showing all 24 units in a single seed. **B.** As in Figure 4FG, but showing all 10 sorted units (columns) for two additional seeds (rows). In addition, the model with only the negative sampling task is shown (green box). **C.** We test a partially observable version of the alternating-T maze where there is no memory of previous observations. Instead, the model has recurrence such that the previous latent state is used to infer the current latent state. **D.** Validation episode score across training for the recurrent model in a partially observable version of the alternating-T maze. Latent size $z = 32$. Displayed is median, with standard error of median for error bars. **E.** As in (D), but with $z = 64$. **F.** As in Figure 4I, but for the recurrent model and partially observable environment used in (C-E).

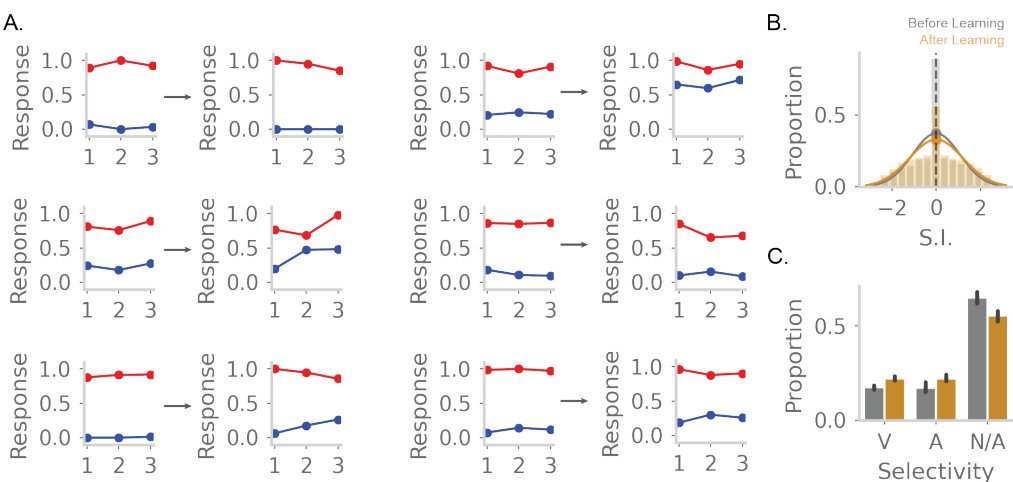

Figure A.6: **A.** As in Figure 5C, but showing six additional units. **B.** As in Figure 5F, but for a model with no value learning head. T-test conducted as in Figure 5F and is not statistically significant (t-statistic: -0.24, p-value: 0.40). **C.** As in Figure 5G, but for a model with no value learning head.