# OpenReview forum: "Predictive auxiliary objectives in deep RL mimic learning in the brain"
_ICLR.cc/2024/Conference — ICLR 2024 oral_

### Official Review · Reviewer_T1AU · 2023-10-25

**Soundness:** 3 good
**Presentation:** 3 good
**Contribution:** 3 good
**Rating:** 8
**Confidence:** 4

**Summary:**

In this work, the authors explore modeling the effect of auxiliary predictive losses on a deep RL agent trained to navigate a spatial gridworld environment. The architecture consists a convolutional encoder that encodes the visual observation, a prediction layer that generates predictions for the auxiliary losses, and a Q-learning agent that takes the encoded state and returns Q-values/actions for the main reward objective. There are two auxiliary losses: a positive sampling loss, which is predicting future states, and negative sampling loss, which encourages representations of non-consecutive states to be distinct from each other. Multiple analyses inspired by neuroscience experiments were employed to show the value of these predictive losses: visualizing the latent state space showing that predictive losses help with representational collapse, showing how long horizon state predictions help with learning new goal locations, looking at how the predictive auxiliary objectives help in producing the "splitting" phenomenon, and showing how the encoder (through virtue of prediction) will adapt to novel transition sequences.

Overall, I think this work is cool and exciting. But I do think portions of the submission have clarity that is below the caliber of an iclr paper. I have initially rated marginally below acceptance (5), but I want to assure the authors that most of my issues are with regards to providing clarity/details. I hope to see the incorporation of more clarity and details in the writing during the rebuttal and would be more than happy to re-evaluate my score after such changes.

**Strengths:**

* The experiments seem to be done soundly and rigorously.

* The authors do an excellent job introducing various relevant neuroscience experiments and grounding the phenomenon into specific predictions for their model.

* The authors use an exciting emerging framework of auxiliary losses to tackle an important problem, which is modeling multiple regions of the brain simultaneously while performing a difficult task.

**Weaknesses:**

The related works section is a little small/sparse. The authors do a good job in highlighting works on auxiliary predictive losses in RL within the machine learning realm, but I think there is also a growing body of work that is using this framework to produce various behavioral phenomenon in cognitive science/neuroscience. These are complimentary works to the current submission and would be good to include. Here are some examples that I feel should definitely be included:

1. Kumar et al. 2022 NeurIPS use auxiliary predictive losses in RL agents to predict abstractions of the observation, operationalized through language and symbolic programs, in order to reproduce abstract human biases.
2. Jensen et al. 2023 bioRxiv introduce a predictive auxiliary loss which helps the agent learn when to plan and reproduces replay patterns seen in rodent hippocampal work.
3. This is not exactly auxiliary predictive loss, but I think it is having the same effect, Binz & Schulz 2022 NeurIPS show adding a regularization term to the loss on the number of bits required to compress the agent's weights reproduces quirks in human exploration.

I think the clarity in some portions in this submission can be improved.

1. On page 4, the description of the auxiliary losses can be improved. Its an important part of the work so it'd be good to have this be as clear as possible. A sentence describing what exactly $\tau$ is representing and why  $\mathcal{L_{+}}$ is a loss term that enforces transition structure in the state representations would be very helpful. Also, for the third term in $\mathcal{L_{+}}$, shouldn't it be $\tau (z_{t+1},a_{t+1})$ and not $\tau (o_{t+1},a_{t+1})$? For the negative sampling loss term, making it explicit $z_{i}$ and $z_{j}$ are latent representations of states that are not consecutive in the same area where the loss term is introduced would be helpful. It'd also be good to have a sentence explaining the motivation for choosing these two specific auxiliary losses. I suspect its loosely inspired by pattern completion vs pattern seperation in hippocampus (respectively) but it would be good to confirm that in this section.

2. It would be helpful to state what the colors in Figure 2D mean. It wasn't clear to me upon first read.

3. The memory component in page 8 is not explained at all. It seems important to describe what this is to put Fig. 4's results in context.

There are a couple of design decisions that left me a little confused (details in the questions section). It would be nice for the authors to clarify the motivations behind them.

Last, I don't think there was any section in the paper explicitly discussing the limitations of the work. I think this is an important part of any iclr paper so it'd be good to include one.

References:
1. Kumar, S., Correa, C. G., Dasgupta, I., Marjieh, R., Hu, M. Y., Hawkins, R., ... & Griffiths, T. (2022). Using natural language and program abstractions to instill human inductive biases in machines. [Advances in Neural Information Processing Systems](https://arxiv.org/abs/2205.11558), 35, 167-180.
2. Jensen, K. T., Hennequin, G., & Mattar, M. G. (2023). A recurrent network model of planning explains hippocampal replay and human behavior. [bioRxiv](https://www.biorxiv.org/content/10.1101/2023.01.16.523429v1.abstract), 2023-01.
3. Binz, M., & Schulz, E. (2022). Modeling human exploration through resource-rational reinforcement learning. [Advances in Neural Information Processing Systems](https://arxiv.org/abs/2201.11817), 35, 31755-31768.

**Questions:**

1. Why use an off-policy learner (Q-learning) rather than on-policy? An on-policy learner would seem to be more biologically realistic. Also I think an actor-critic approach (e.g. A2C, PPO, etc) may reflect what striatum is doing more than Q-learning?

2. Is there a reason why there is no recurrence in the model? Striatum will indirectly project back to sensory cortex via thalamus. Also it may be possible that hippocampus projects back to sensory visual cortex. Hippocampal activity early in a trial can be predictive of information in visual cortex at a later time in the trial (see Hindy et al. 2016 Nature Neuroscience). Regardless of the biological realism of these recurrent connections, I think there could be useful normative principles in having recurrent loops between modules in this architecture.

3. Fig 5c: The change in firing rates seem pretty small between before/after exposure. Are these changes statistically significant?

References:
Hindy, N. C., Ng, F. Y., & Turk-Browne, N. B. (2016). Linking pattern completion in the hippocampus to predictive coding in visual cortex. [Nature neuroscience](https://www.nature.com/articles/nn.4284), 19(5), 665-667.

---

> ### Author Response · Authors · 2023-11-21
> **Response to Reviewer T1AU (Part 1)**
>
> Thank you very much for your considered, thoughtful, and very actionable review. We have made several modifications to the manuscript, including adding exposition, citations, and new experiments, with the intention of addressing the concerns raised in this review. These include
>
> * Expanding the related work, in particular incorporating the recommended citations
> * Adding clarifications about modeling choices in the methods section and about experiment design in the results section
> * Adding a limitations paragraph in the discussion
> * Adding an additional experiment testing the effects of recurrency in the model in a partially observable task
> * Adding statistical significance tests to the Li & DiCarlo experiments of Figure 4.
>
> We describe these in more detail below, and respond specifically to your comments. We are furthermore happy to respond further if there are any more questions.
>
> Weaknesses:
> > The related works section is a little small/sparse... there is also a growing body of work that is using this framework to produce various behavioral phenomenon in cognitive science/neuroscience. These are complimentary works to the current submission and would be good to include
>
>  Thank you for all the suggestions! We have updated the related works section with the reviewer’s suggestions, as well as additional references that help contextualize the work.
>
> “A growing body of work considers modular and multi-objective approaches to building integrative models of brain function. One approach has been to construct multi-region models by combining modules performing independent computations and comparing representations in these models to neural activity (Frank & Claus, 2006; O’Reilly & Frank, 2006; Geerts et al., 2020; Russo et al., 2020; Liu et al., 2023, Jensen et al. 2023). On the behavioral end, there has also been prior work discussing how the addition of biologically-realistic regularizers or auxiliary objectives can result in performance more consistent with humans (Kumar et al., 2022; Binz & Schulz, 2022; Jensen et al., 2023). Our work differs in that the entire system consists of a neural network that is trained end-to-end, allowing us the opportunity to specifically study the effects on representation learning. In this paper, we show how deep RL networks can be a testbed for studying representational changes and serve as a multi-region model for neuroscience.”
>
> > On page 4, the description of the auxiliary losses can be improved. It’s an important part of the work so it'd be good to have this be as clear as possible. A sentence describing what exactly \tau is representing and why L_+ is a loss term that enforces transition structure in the state representations would be very helpful.
>
> Thank you for the suggestion! We’ve added the following clarification in section 3 where we describe the role of $L_+$:
>
> “The positive sample loss is defined as $L_+ = ||\tau(z_t, a_t) - z_{t+1} - \gamma \tau(z_{t+1}, a_{t+1})||^2$, where $z_{t} = E(o_{t})$ and $\tau(z_{t}, a_{t}) = z_{t} + T(z_{t}, a_{t})$. That is, in the $\gamma=0$ case, the network $T$ is learning the difference between current and future latent states such that $\tau(z_t, a_t)=z_t + T(z_t,a_t) \approx z_{t+1}$. This encourages the learned representations $z$ to be structured so as to be consistent with predictable transitions (Francois-Lavet et al, 2019).”
>
> > Also, for the third term in L_+, shouldn't it be $\tau(z_{t+1}, a_{t+1})$ and not $\tau(o_{t+1}, a_{t+1})$?
>
> Yes, thanks for the catch! It is now fixed.
>
> > For the negative sampling loss term, making it explicit z_i and z_j  are latent representations of states that are not consecutive in the same area where the loss term is introduced would be helpful.
>
> Thanks for the suggestion! We’ve added the following in the beginning of paragraph 3 of section 3:
> “We emphasize that $z_i$ and $z_j$ are randomly sampled from the buffer and thus may represent states that are spatially far from another.“
>
> > It'd also be good to have a sentence explaining the motivation for choosing these two specific auxiliary losses. I suspect it's loosely inspired by pattern completion vs pattern separation in hippocampus (respectively) but it would be good to confirm that in this section.
>
> Thanks for the suggestion! That is part of the motivation. We have expanded the explanation in section 3 to the following:
>
> “This loss drives temporally distant observations to be represented differently, thereby preventing the trivial solution from being learned (mapping all latent states to a single point). The use of two contrasting terms ($L_-$ and $L_+$) is not just useful for optimization reasons -- it also mirrors the hypothesized pattern separation and pattern completion within the hippocampus (O’Reilly & McClelland, 1994; Schapiro et al., 2017). However, we note that negative sampling elements are not always needed to support self-predictive learning if certain conditions are satisfied (Tang et al 2023).”

---

> > ### Author Response · Authors · 2023-11-21
> > **Response to Reviewer T1AU (Part 2)**
> >
> > > It would be helpful to state what the colors in Figure 2D mean. It wasn't clear to me upon first read.
> >
> > We add the following sentence in the caption for Figure 2D:
> > “Diagram of the encoder network (red), learned latent state (gray), and value-learning network (blue).”
> >
> > > The memory component in page 8 is not explained at all. It seems important to describe what this is to put Fig. 4's results in context.
> >
> > We have modified the description to clarify this:
> > “That is, the input into the encoder at time $t$ is $o_t + \alpha o_{t-1} + \alpha^2 o_{t-2}+\dots$ for some $\alpha<1$. This decaying sum of recent observations captures information about the recent past in a simple way, and is inspired by representations hypothesized by temporal context model (Howard & Kahana, 2002). “
> >
> > > Last, I don't think there was any section in the paper explicitly discussing the limitations of the work. I think this is an important part of any iclr paper so it'd be good to include one.
> >
> > Thank you for this suggestion – we agree it is a good practice. We add the following sentence to the conclusion:
> > “Our results are limited in the complexity of tasks and the diversity of auxiliary objectives tested. Future work can improve on current understanding by more systematically comparing effects across objectives over more complex tasks. We also did not examine representations developed in the value learning network, which is ripe for comparison with striatum data.”
> >
> > **Questions:**
> >
> > > Why use an off-policy learner (Q-learning) rather than on-policy? An on-policy learner would seem to be more biologically realistic. Also I think an actor-critic approach (e.g. A2C, PPO, etc) may reflect what striatum is doing more than Q-learning?
> >
> > This is a really great question! It would be interesting to explore how different value learning networks affect the robustness of the results and the representations found in the system. We unfortunately did not have time to test these different simulations. There were a few factors that motivated our initial  design choice to use a Q-learning system:
> > 1. There is some precedent in that previous papers model the striatum with Q-learning and discuss experiments that indicate the striatum may be explained by both actor-critic and Q-learning setups (Geerts & Burgess 2020, Averbeck & O’Doherty 2022, Blackwell & Doya 2023, ). For our purposes, using a deep Q network was a simple approximation for a general model-free reinforcement learner. Interestingly, a recent theoretical paper has suggested the presence of action-surprise signals in striatal dopamine combined with an actor-critic architecture results in a model that approximates Q-learning (Lindsey & Litwin-Kumar 2022).
> > 2. Freely behaving animals have been observed to learn in an offline manner. For instance, Rosenberg & Meister 2021 show that freely behaving mice in a maze will behave largely randomly in initial environment explorations. These mice will then make directed and efficient trajectories to rewards when not exploring.
> > 3. Finally, the offline learning setting is relevant because it connects to the hippocampus in many ways. Hippocampal replay has been suggested to contribute to offline RL as the supplier of the replay buffer used in the RL updates (Mattar & Daw 2018). The negative sampling we used in the predictive model (which relies on random replay) is also supported by hypotheses of pattern separation in the hippocampus. Thus, there are interesting biological similarities between the use of an offline replay buffer and activity in the hippocampus. In future experiments, we are interested in using this structure to test replay-related hypotheses.

---

> ### Author Response · Authors · 2023-11-21
> **Response to Reviewer T1AU (Part 3)**
>
> **Questions (continued)**
> > Is there a reason why there is no recurrence in the model? Striatum will indirectly project back to sensory cortex via thalamus. Also it may be possible that hippocampus projects back to sensory visual cortex. Hippocampal activity early in a trial can be predictive of information in visual cortex at a later time in the trial (see Hindy et al. 2016 Nature Neuroscience). Regardless of the biological realism of these recurrent connections, I think there could be useful normative principles in having recurrent loops between modules in this architecture.
>
> We agree this is an important direction to investigate, and we have added experiments to the manuscript that speak to it. To begin testing this, we added recurrency in the model and tested the model on a partially observable version of the alternating-T maze task in Figure 4. These new results are added to Appendix Figure 5. In this version, the model is only provided with a 2D visual observation that indicates its current location ($o_t$). The model is modified with recurrency such that its previous latent state ($z_{t-1}$) is fed back into the encoder. Thus, $z_t = E(o_t, z_{t-1})$. To solve the task, the model must use recurrency to maintain an internal memory of its current context. We find that the predictive auxiliary objective greatly improves the model’s ability to learn this partially observable task (Figure A5DE). The splitting across population representations are still similar as in the fully observable case (compare Figure A5F with Figure 4I). These experiments show how recurrency can be used to solve more complex tasks. We also found from these experiments that learning can still be stable even in a model with both recurrency and self-predictive objectives. Finally, the representations that resulted from these experiments show how recurrency strongly biases representations to be more ‘split’/decorrelated across conditions.
>
> However, we still have not explored the effects of recurrency across modules. As the reviewer points out, there is plenty of experimental and anatomical evidence to suggest that recurrency plays a strong effect across brain areas. For simplicity and ease of interpretability, we started with a feedforward network. There is certainly a lot of room to improve on this aspect of the model. We mention this as a useful direction in the discussion paragraph discussing limitations of our work.
>
> > Fig 5c: The change in firing rates seem pretty small between before/after exposure. Are these changes statistically significant?
>
> To address this comment, we ran a t-test across samples of firing rate changes before/after exposure (i.e. the values displayed in Figure 5D). We find that the change in firing rate of both models (the model with a predictive objective, and the model without a predictive objective) are significantly different than 0:
>
> | Model | sequence location | p-value | t-statistic |
> | :-: | :-: | :-: | :-: |
> | With Prediction | 1 | $0.002$ | $-3.69$ |
> | With Prediction | 2 | $3.2 \times 10^{-7}$ | $-9.04$ |
> | With Prediction | 3 | $7.4 \times 10^{-5}$ | $-5.5$ |
> | Without Prediction | 1 | $0.0009$ | $-4.17$ |
> | Without Prediction | 2 | $0.002$ | $-3.74$ |
> | Without Prediction | 3 | $0.009$ | $-3$ |
>
> In addition, we also run a two-sample t-test to check how different the means of both models are from each other:
>
> | Sequence location | p-value | t-statistic |
> | :-: | :-: | :-: |
> | 1 | $0.48$ | $0.71$ |
> | 2 | $0.03$ | $2.32$ |
> | 3 | $0.03$ | $2.24$ |
>
> Thus, we conclude that the model with predictive objective does change the firing rates to flip the preference of neurons before/after exposure. In particular, the effect of this flip is strongest at the images close to the swap location (e.g., sequence locations 3 and 2), consistent with the observations in the Li & DiCarlo papers. We add annotations regarding significance into Figure 5D and the caption for Figure 5D.

---

> ### Comment · Reviewer_T1AU · 2023-11-21
>
> Thank you for the detailed response! This addresses all my concerns; I have raised my score accordingly.
>
> Hope you guys have a happy Thanksgiving! (if you're in a place in the world that celebrates it)

---

### Official Review · Reviewer_CL1C · 2023-10-30

**Soundness:** 4 excellent
**Presentation:** 4 excellent
**Contribution:** 3 good
**Rating:** 8
**Confidence:** 4

**Summary:**

This paper implements a deep RL framework with predictive auxiliary objectives for representation learning. The authors demonstrate in a gridworld setting that predictive objectives improve representation learning by preventing representation collapse and enhancing transfer learning when transition structure remains unchanged. The paper also relates the components of the deep RL model to different brain regions, including the sensory cortex, the hippocampus, and the striatum. They show that representation learning in the predictive model resembles the neural activity observed in the brain, and learning in the encoder model resembles neural observations in the visual cortex.

**Strengths:**

- Overall I really enjoy reading this work, due to its clear presentation both in the text and in the figures, experiments testing different perspectives of the model, and the strong link to the brain
- This work introduces a multi-region model that is developed from a normative perspective, instead of fitting to recorded data, which can be extended to other tasks and to test against new biological evidence
- I appreciate the discussion of the limitation that predictive auxiliary objectives may be less helpful when the transition structure or policy changes

**Weaknesses:**

- It's interesting to see in section 4.4 where the authors describe the effects of value learning in the encoder network, but this part feels somewhat disconnected from the rest of the paper, as the primary focus is to demonstrate how predictive objectives can lead to representation changes similar to those seen in the brain

**Questions:**

- I'm curious to learn if there is a similarity between the action selection network and the neural activity observed in the striatum
- Figure 2: Were cells that didn't show place-like activities filtered out in these analyses?
- Were there also place-like activities in the encoder model?

---

> ### Author Response · Authors · 2023-11-21
> **Response to Reviewer CL1C**
>
> Thank you very much for your thoughtful and positive review! We have addressed your questions and comments below.
>
> > It's interesting to see in section 4.4 where the authors describe the effects of value learning in the encoder network, but this part feels somewhat disconnected from the rest of the paper, as the primary focus is to demonstrate how predictive objectives can lead to representation changes similar to those seen in the brain
>
> Thanks for noting the lack of clarity! We were not just interested in the predictive network but also wanted to explore how the other modules in the system mutually interacted. We make a few edits to the introduction to emphasize that we are interested in the roles of both learning heads.
>
> In paragraph 2 of the introduction we add:
> “It is unclear how value learning, predictive objectives, and feature learning mutually interact to shape representations.”
>
> In the final paragraph of the introduction we make a few modifications:
> “We further demonstrate that a deep RL model with multiple objectives undergo a variety of representational phenomena also observed in neural populations in the brain. Downstream objectives can alter activity in the encoder, which is mirrored in various results that show how visual cortical activity is altered by both predictive and value learning. Additionally, learning in the prediction module drives activity patterns consistent with activity measured in hippocampus.”
>
>
> **Questions**
> > I'm curious to learn if there is a similarity between the action selection network and the neural activity observed in the striatum
>
> This is a great question! There is prior work that has explored the similarities between action selection networks in deep RL models and striatal neural activity. For instance, Dabney & Botivinick 2020 show how signals in the ventral tegmental area, a main projector to the striatum, can be explained by value encoding in the distributional RL framework. Lowet & Uchida, in an upcoming 2023 NeurIPs UniReps workshop paper, further this line of work by recording from striatum and demonstrating consistency with a distributional RL model as well. As another example, Lindsey & Litwin-Kumar 2022 show how action-modulated striatal dopamine responses can be explained by value encoding in a deep actor-critic model that is approximating deep Q-learning.
>
> However, there is still much to explore in comparing the value network with striatal experiments. In particular, there are likely interesting effects resulting from the other modules of the system, and our model would be a great testbed for these effects. Although we did not have the time to explore this, we added a sentence in the discussion paragraph discussing limitations of our work: “We also did not examine representations in the value learning network, which is ripe for comparison with striatum data.”
>
>
> > Figure 2: Were cells that didn't show place-like activities filtered out in these analyses?
>
> All cells were included in population level analyses, without any exclusion criteria. However, to display example place fields for Figure 4B, cells are sorted by spatial modulation and the top four cells are shown for each cell. All additional cells are pictured in the Appendix, however.
>
>
> > Were there also place-like activities in the encoder model?
>
> This is a great question! We do find place-like activity in the encoder model. We think there are two interesting points of discussion around this finding:
>
> 1. Place-like activity has been found in various sensory areas (Fiser & Keller 2016, Town & Bizley 2017, Long & Zhong 2021). Thus, perhaps the fact that our encoder network shows place-like activity provides a possible explanation for this finding in the experimental literature. That is, the combination of spatially correlated inputs and downstream predictive learning can induce spatial structure into upstream areas.
>
> 2. On the other hand, it should be noted that we optimized our network for one task, and optimized all networks end-to-end. It's unclear if such place-like representations would arise under a more realistic setup. For instance, the encoder network could be asked to learn  representations for many tasks, some of which are not spatial. This may result in encoder representations more agnostic to space. As another example, the learning rate of the encoder network could be slower than that of the downstream hippocampus-like network (Tang et al). This could result in the spatial structure being mostly stored in the hippocampus-like network and not being propagated to upstream regions like the encoder.

---

### Official Review · Reviewer_f2kS · 2023-11-01

**Soundness:** 3 good
**Presentation:** 3 good
**Contribution:** 3 good
**Rating:** 8
**Confidence:** 3

**Summary:**

The authors use a multi-objective RL model that combines a Q-learning module with an auxiliary predictive objective to demonstrate advantages of predictive learning on representations for multi-regional interactions in neural networks.  Using foraging tasks on gridworld scenarios as an example, they demonstrate that using an auxiliary predictive module results faster training and smoother representations of the task environment.  They also demonstrate that increasing the time horizon for predictive learning produces networks that retrain faster on new tasks in the same environment.  Interestingly, the learned representation is crucial for this as scrambling the transition structure results in much slower learning. They then demonstrate that some of the representational changes observed in the model reflect similar changes observed in real neural networks.

**Strengths:**

The description of the methods and approach is fairly clear and the overall goals of the paper are clear, with some room for improvement.  The numerical experiments provided demonstrate how a predictive loss benefits the learned representations available in a downstream area (not necessarily directly related to the area responsible for prediction), without the "predictive area" necessarily providing any direct information to the area responsible for valuation and action selection.

**Weaknesses:**

One important feature that isn't clear from what is presented in the paper is how the environment itself may or may not affect these results.  I don't find any examples of what gridworld environments are being solved by these models, let alone how complex they are.  Surely the predictability of the environment itself has some bearing on the rate of learning and retrainability for novel tasks, not to mention the quality of representations?  Please provide examples of these environments.  If possible, please consider varying the complexity of the environments.

minor:

- Bottom of page 3 "the standard double deep Q-learning temporal difference loss function":  Even though standard, please either provide the form of this loss or provide a reference.

- page 4, just below figure caption, definition of positive sample loss:  should o_{t+1} be z_{t+1}?

- last sentence, first paragraph page 5:  "the predictive model is trained with..."  (No "be")

- first sentence, section 4.2:  remove "is used" at the end of the sentence.

- first sentence, last paragraph on page 7:  remove "to",I.e. "undergo experience-dependent changes".

- last paragraph, page 8:  "remembering the previous trial type". (Remove "whether"), also empty reference at the end of that sentence.

**Questions:**

- Can you provide example environments?

- What is the effect of varying the complexity of the environment?


Similar to the above, the encoder seems to have all available information about the environment (in principle), so the predictive task is in some sense simpler than it otherwise might be for a real organism, which only has clues to its environment.  Do you have thoughts about how partial observability might affect the predictive module?  (This is beyond the scope of the paper, but might be worth speculating on.)

---

> ### Author Response · Authors · 2023-11-21
> **Response to Reviewer f2kS (Part 1)**
>
> Thank you very much for your considered, thoughtful review. We have now added a number of experiments with the intention of addressing the concerns mentioned in this review, which we believe have improved the manuscript. These include:
> * experiments comparing results in a with stochastic environment
> * experiments replicating results in a more complex CIFAR environment
> * experiments in a partially observable environment
>
> We describe these in more detail below, and respond specifically to your comments. We are furthermore happy to respond further if there are any more questions.
>
> > I don't find any examples of what gridworld environments are being solved by these models, let alone how complex they are.
>
> To make this more clear, we have now included examples of the environment and corresponding visual inputs into Appendix Figure 2. We use a gridworld environment where the underlying state space is a 8x8 grid with walls at the edges of the arena. Four discrete actions are possible (up, down left, right) and the 2D  visual inputs to the agent are either (A) a top-down view of the agent’s location in the arena or (B) the same top-down view with pixels randomly shuffled. In recently introduced experiments (described below) we also experiment with (C) CIFAR images as visual input for added visual complexity. In case B and C, this removes spatial correlations between visual inputs from neighboring states.
>
> > Surely the predictability of the environment itself has some bearing on the rate of learning and retrainability for novel tasks, not to mention the quality of representations? Please provide examples of these environments.
>
> Yes, this is an important point. The predictive auxiliary objective assumes that the effects of actions in the environment are predictable. Specifically, the predictive network is trained to predict $z_{t+1}-z_{t}$ given $(z_t, a_t)$.
>
> To further test the effects of environment predictability, we ran additional experiments in a stochastic gridworld environment and included these results in Appendix Figure 3. In this environment, transitions are not deterministic. There is a probability $p$ that the agent’s action $a$ is not obeyed and instead the environment transitions according to an unselected action instead. We find that, as $p$ increases, the relative benefit of the predictive objective vanishes and all models perform similarly (Figure A3D). We discuss these results in the bottom of section 4.1.

---

> ### Author Response · Authors · 2023-11-21
> **Response to Reviewer f2kS (Part 2)**
>
> > If possible, please consider varying the complexity of the environments... What is the effect of varying the complexity of the environment?
>
> We agree that varying the complexity of the environment would be a helpful way to test the robustness of the observed effects.
>
> To further test the effects of environment complexity, we ran a new set of experiments in which the observation at each state in the gridworld environment consisted of a different CIFAR image, and included these results in Appendix Figure 2-3. We find the results to be consistent with the simpler gridworld inputs we use in the main figures (Figure A3C). That is, predictive auxiliary objectives enable faster learning of the gridworld task compared to other models (Figure A3C). Additionally, the model equipped with a predictive objective can learn the task for small latent sizes whereas the model without auxiliary objectives cannot (Figure A3C). We update section 4.1 to discuss these results.
>
> We also note that prior work has tested predictive auxiliary objectives in single-task settings with greater task complexity. In these findings, the inclusion of these objectives consistently improves performance over models without auxiliary objectives (Jaderberg et al., 2016; Shelhamer et al., 2016; Shelhamer et al., 2016; Oord et al., 2018; Wayne et al., 2018).
>
> > Do you have thoughts about how partial observability might affect the predictive module?
>
> We predict that a predictive auxiliary objective may provide additional benefits in a partially observable task. For instance, the additional predictive information induced by the transition module can help the network infer hidden state dynamics better. In cases of observation noise, predictive information can also help network representations ignore irrelevant or noisy parts of the observation.
>
> To further test this, we ran new experiments with a partially observable version of the alternating-T maze task in Figure 4H. These new results are added to Appendix Figure 5. In this version, the model is only provided with a 2D visual observation that indicates its current location ($o_t$). The model is modified with recurrency such that its previous latent state ($z_{t-1}$) is fed back into the encoder and can be used to infer the current latent state. Thus, $z_t = E(o_t, z_{t-1})$. To solve the task, the model must use recurrency to maintain an internal memory of its current context. We find that the predictive auxiliary objective greatly improves the model’s ability to learn this partially observable task (Figure A5D). The splitting across population representations are still similar as in the fully observable case ( Figure A5F vs Figure 4I). We discuss these results in the bottom of section 4.3.
>
> **Minor comments**
>
> We implemented all changes suggested in the minor comments.

---

> > ### Comment · Reviewer_f2kS · 2023-11-23
> >
> > Thank you for the responses.  I am raising my score.

---

### Author Response · Authors · 2023-11-23
**Response to All Reviewers**

We want to thank the reviewers for their thoughtful, constructive, and largely positive reviews! To summarize, in response to reviewer comments, we have made a number of modifications to the experiment that we believe improve the manuscript. These are:

- Experiments in a visually complex environment in which observations consist of CIFAR images, which show that the observed effects generalize to a more complex, larger scale setup.
- Experiments exploring the addition of recurrence to our model architecture on partially observable environments, showing that the predictive objective is especially helpful
- Experiments evaluating performance in a stochastic environment
- Reporting statistical significance
- Clarifications in exposition
- Expanding the related work section.

We again thank the reviewers and the AC for their time.

---

### Meta-Review · Area_Chair_RshH · 2023-12-24

**Metareview:**

This paper presents a multi-objective RL model that combines a Q-learning module with an auxiliary predictive objective to demonstrate advantages of predictive learning on representations for multi-regional interactions in neural networks.
The authors evaluate the the effect of auxiliary predictive losses on a deep RL agent. The architecture consists a convolutional encoder that encodes the visual observation, a prediction layer that generates predictions for the auxiliary losses, and a Q-learning agent that takes the encoded state and returns Q-values/actions for the main reward objective. There are two auxiliary losses: a positive sampling loss, which is predicting future states, and negative sampling loss, which encourages representations of non-consecutive states to be distinct from each other.

Strength and weaknesses:
+ All reviewers agreed on clarity of the presentation
+ Also worth noting is the thoroughness of the experiments
+ Surprising conclusion: "Learning predictions is sufficient to induce useful structure into representations used by other regions."

- Experimental complexity is limited, which may result in doubts in extending the claims to scalable DeepRL models in larger/more involved domains.

**Justification For Why Not Higher Score:**

Reviewers provide a high score, but limited experimental complexity limits the scope of claims.

**Justification For Why Not Lower Score:**

All Reviewers are in agreement of the novelty and technical value of the method.

---

### Decision · Program_Chairs · 2024-01-16

Accept (oral)